# Experimental Study of Recycled Concrete under Freeze–Thaw Conditions

**DOI:** 10.3390/ma17163934

**Published:** 2024-08-08

**Authors:** Alipujiang Jierula, Cong Wu, Zhixuan Fu, Hushitaer Niyazi, Haodong Li

**Affiliations:** 1College of Architecture and Engineering, Xinjiang University, Urumqi 830046, China; congwu@stu.xju.edu.cn (C.W.); huxtarniyaz@163.com (H.N.); lhd_0381@163.com (H.L.); 2Xinjiang Key Laboratory of Building Structure and Earthquake Resistance, Xinjiang University, Urumqi 830046, China; 3Graduate School of Education, University of Melbourne, Melbourne, VIC 3000, Australia; zhixuanfu@student.unimelb.edu.au

**Keywords:** recycled concrete, freeze–thaw cycles, durability, compressive strength, aggregate replacement

## Abstract

Recycled concrete is a new and environmentally friendly material for future construction. When applying recycled concrete to cold and severe regions, it is necessary to consider the freeze–thaw resistance of recycled concrete. Using an indoor freeze–thaw cycle test system, the rapid freezing method was used to conduct rapid freeze–thaw tests on recycled concrete specimens under set freeze–thaw cycle conditions. Relevant parameters (such as compressive strength, quality loss rate, rebound value) were tested on recycled concrete specimens that completed the set number of freeze–thaw cycles. The influence of various factors (freeze–thaw cycle number, replacement rate of recycled aggregates) on the compressive strength, quality loss rate, and rebound value of recycled concrete was analyzed. The results indicate that with the increase of freeze–thaw cycles and recycled aggregate content, the workability, rebound value, and compressive strength of the specimens decrease, and the quality loss rate increases. The changes in workability of concrete are sharp, with a slump difference of 21 mm between concrete with 100% recycled coarse aggregate and concrete using natural coarse aggregate. The rebound value of the specimen shows a decreasing trend overall, but the rebound value varies for different measuring points on the same specimen. The attenuation of compressive strength is significant. When fully using recycled aggregates to prepare concrete, the compressive strength after 30 freeze–thaw cycles decreases by 49.42% compared to the compressive strength after 0 freeze–thaw cycles. The overall quality loss rate shows a decreasing trend, but the quality loss is not severe.

## 1. Introduction

The construction industry has embraced green, energy efficient, renewable practices, while urbanization has led to pollution and resource depletion issues. Concrete is a vital component in construction, with global demand projected to reach 5 billion tons annually by 2050 [1]. When concrete reaches its service life, how to dispose of the waste concrete is a big problem. These piles of construction wastes can consume land resources and pollute water and soil. There are many sources of waste concrete, such as aging, explosions, earthquakes, floods, and typhoons. It is very necessary to reuse the solid waste from construction. Solid waste concrete has significant environmental, economic, and sustainable benefits [2]. The resource utilization of construction solid waste will reduce the discharge of solid waste, occupy less land area, and reduce the destructive power on soil. Another benefit of the resource utilization of construction solid waste is the reduction of emissions of gases such as CO_2_. A total of 25% to 50% of the CO_2_ emitted during cement manufacturing can be reabsorbed by the resource utilization process of waste concrete. Compared with stacking or landfilling methods, it can reduce N_2_O emissions by 50%, SO_2_ emissions by 30%, CO emissions by 28%, and CO_2_ emissions by 10%. Recycled aggregates produced from solid waste also have good mechanical properties, which can meet the demand for aggregates in the construction industry [3].

Since the 1950s, developed countries such as Germany, the United States, and Japan have recognized the importance of building solid waste resource utilization and have begun to explore ways and methods of building solid waste resource utilization, making it a sustainable development goal.

Germany is one of the earliest countries to legislate on construction solid waste. After launching the “Blue Angel” plan in 1978, it formulated regulations such as the “Waste Management Law” and later promulgated the “Circular Economy and Waste Removal Law”, which had a wide impact on the development of the world’s construction solid waste resource utilization. In addition to legislation, Germany has also taken reasonable economic measures to guide the treatment of construction solid waste. The treatment methods for construction solid waste in Germany mainly include landfill and recycling. From the perspective of production technology, Germany currently has the ability to process construction solid waste on a large scale. For example, in 2008, Germany produced approximately 60 million tons of recycled aggregate, accounting for 10.6% of the total aggregate. The world’s largest construction solid waste treatment plant is located in Germany, capable of producing 1200 tons of recycled construction solid waste materials per hour. As of 2020, there are about 200 construction solid waste enterprises in Germany with an annual turnover of 2 billion euros. Recycled materials not only save a large amount of natural materials, but also avoid the social problems caused by the accumulation of construction solid waste.

The United States is one of the major countries in urban waste generation, producing up to 800 million tons of urban waste annually, of which construction solid waste accounts for over 40%. The resource utilization of construction solid waste in the United States started earlier and has formed an independent system that is suitable for the national conditions in terms of regulations, policies, and practical applications. The United States has achieved a 70% recycling rate of construction solid waste through the application of resource utilization technology, with the remaining 30% being treated through a centralized landfill.

Japan has a small land area and scarce material resources, so it pays more attention to the sustainable use of resources. In Japan, construction solid waste is referred to as “building auxiliary products” and is mainly divided into two types: renewable resources and waste. From the perspective of treatment effectiveness, the recycling rate of solid waste in Japanese construction reached 65% in 1995, and this indicator rose to 81% in 2000. A survey in 2007 showed that the total amount of waste produced by construction sites in Japan was 63.8 million tons, of which only 4.02 million tons were ultimately treated as garbage, with a recycling rate of 92.2%. Japan has very strict requirements for garbage classification and has designed and produced a complete set of construction solid waste equipment with high sorting rates, which can complete crushing, separation, screening, and even radiation monitoring.

South Korea is one of the early Asian countries to carry out the resource utilization of construction solid waste. After long-term efforts, it has basically achieved the resource utilization of construction solid waste. As early as 2008, there were 337 construction solid waste treatment enterprises in South Korea. In 2002, the amount of construction solid waste in South Korea was about 120,000 tons, while the amount of resource utilization was about 100,000 tons, with a resource utilization rate of 83%. In 1998, recycled aggregates accounted for 76% of construction solid waste, but by 2006, this value had reached as high as 97%.

Many countries are vigorously promoting the resource utilization of construction solid waste, but there are still many problems to be solved in its application. In cold regions, concrete materials are susceptible to freeze–thaw cycles, resulting in frequent freeze–thaw damage, deterioration of concrete performance, and even reduced lifespan, leading to the destruction and collapse of buildings [4]. Scholars have been investigating the freeze–thaw cycle failure mechanism of concrete as early as 80–90 years ago and have since obtained a series of academic achievements, including: (1) hydrostatic pressure theory [5,6,7], (2) osmotic pressure theory [8,9], (3) Litvan theory [10,11], (4) critical water saturation theory [12,13], (5) thermodynamic theory [14], (6) micro ice lens model theory [15], (7) qualitative continuous damage model theory [16], and (8) Muru theory [17]. In contrast, the frost resistance of recycled concrete is related to many factors such as the source of primary concrete, water saturation, replacement rate of recycled coarse aggregates, and preparation process. Therefore, there is currently no clear mechanism for the freeze–thaw degradation of recycled concrete.

The frost resistance of recycled concrete is crucial, determining its applicability in cold regions. Many scholars have conducted relevant research. The results of Zaharieva et al. [18] showed that recycled concrete may cause severe freeze–thaw damage when exposed to freezing conditions. This is because it drains water into the surrounding cement mortar. The study concludes that recycled concrete can only be used in moderately cold regions if the water–cement ratio is less than 0.55. In severe cold regions, salty soil is often present. Su et al. [19] revealed that the mass of recycled concrete increases and decreases with the number of salt freezing cycles. Additionally, the compressive strength and modulus of elasticity of recycled concrete decrease with the increase in salt freezing cycles. Lotf et al. [20] explored why the frost resistance of recycled concrete is inferior to that of ordinary concrete and found that bonded cement mortar is the reason for the poor frost resistance of recycled concrete. The analysis of frost resistance conducted by Omary et al. [21] revealed that the frost resistance of recycled concrete is inferior to that of natural concrete. The compressive strength of recycled concrete depends on the Los Angeles coefficient of gravel. The number of freeze–thaw cycles can reflect the intensity of the recycled concrete subjected to freeze–thaw damage. The experimental conclusion of Wei et al. [22] revealed that with the increase of freeze–thaw cycles, the compressive strength of the recycled concrete gradually decreases. The pores of the recycled concrete are the first to increase and then decrease. 

According to scholarly research, the strength of recycled concrete may be weakened after undergoing cycles of freezing and thawing. Several scholars have quantified the magnitude of the strength reduction. The experimental findings of Buck et al. [23] indicate that the strength reduction can range from 5% to 42% after freeze–thawing cycles. Scholars have explored the reasons behind this decrease in strength. Salem et al. [24] discovered that the highly hygroscopic nature of recycled aggregate in recycled concrete intensifies the freezing effect in freeze–thaw environments. However, some experts suggest that recycled concrete may exhibit comparable or superior frost resistance to natural concrete. Yildirim et al. [25] conducted a study involving 300 freeze–thaw cycles on recycled concrete specimens. They found that if the replacement rate of recycled concrete is less than 50%, then frost resistance of recycled concrete is comparable to that of natural concrete. Cao et al. [26] stated that if recycled concrete is used for coarse aggregate and natural gravel is used for fine aggregate and the replacement rate of recycled aggregate is less than 50%, the mechanical properties of recycled concrete and natural concrete are similar. Fan et al. [27] discovered that the frost resistance of recycled concrete is optimal and similar to that of natural concrete when the replacement rate of recycled aggregate is 33%. Thomas et al. [28] concluded that recycled aggregate has less impact on the properties of concrete when used at the same water–cement ratio. Ohemeng et al. [29] concluded that recycled aggregates made from high-strength concrete with a strength greater than 50 Mpa generally do not negatively affect the strength properties of recycled concrete.

According to the literature review above, there are some shortcomings in the research on recycled concrete. 1. There is no clear mechanism for freeze–thaw damage. 2. Most studies only characterize frost resistance through compressive strength.

This article aims to study the frost resistance of concrete with different amounts of recycled coarse aggregate. And the frost resistance of recycled concrete is characterized by parameters such as rebound value, quality damage rate, compressive strength, etc. Meanwhile, an explanation was provided for the freeze–thaw damage mechanism of concrete with added recycled coarse aggregates. Finally, some methods to enhance the mechanical properties of recycled concrete are presented.

The experiment adopts the fast-freezing method to conduct rapid freeze–thaw tests on specimens with different replacement rates of recycled coarse aggregates (0%, 25%, 50%, 75%, 100%). The compressive strength, mass loss rate, rebound value, and workability of the specimens under different freeze–thaw cycles (0, 15, 30) are observed, revealing the freeze–thaw rupture mechanism of recycled concrete and providing a theoretical basis for the application of recycled concrete in harsh and cold regions. 

According to the experimental results, we concluded that when using recycled concrete in cold regions, attention should be paid to minimizing the amount of waste mortar on the surface of recycled aggregates. The replacement rate of recycled coarse aggregates can be selected according to actual strength needs, but good performance can be achieved when it does not exceed 50%.

## 2. Purpose and Scope of the Research

The purpose of this study is to investigate the effect of the dosage of recycled coarse aggregate on the frost resistance of concrete, and to find the optimal dosage of recycled coarse aggregate in cold regions to guide practical engineering construction. The reason for doing these tasks is that with the increasing application of recycled aggregates in concrete, it is necessary to understand the impact of recycled aggregates on concrete. When using recycled aggregates in cold regions, the amount of recycled coarse aggregates added is crucial. Many scholars have studied the influence of freeze–thaw environment on the strength and durability of recycled concrete but have not addressed the effect of recycled coarse aggregates on the frost resistance of concrete. In order to investigate the effect of recycled coarse aggregate content on the frost resistance of concrete, recycled coarse aggregates with replacement rates of 0%, 25%, 50%, 75%, and 100% were selected. This is beneficial for us to comprehensively observe the influence of replacement rates on the frost resistance of concrete. Secondly, due to the significantly lower mechanical properties of recycled concrete compared to ordinary concrete and the severe damage caused by excessive freeze–thaw cycles, it is not conducive to analyzing the impact of recycled aggregates on concrete. Therefore, 15 freeze–thaw cycles and 30 freeze–thaw cycles were selected. We tested the compressive strength, rebound value, and quality loss rate of concrete with different amounts of recycled aggregate by freeze–thaw cycles. We have obtained the optimal dosage of regenerated rough bone material for guiding cold regions.

## 3. Experimental Program

### 3.1. Material

Cement: P-O42.5R grade cement produced by Xinjiang Tianshan Cement Co., Ltd. (Urumqi, China), main chemical composition, see Table 1.

Aggregates: particle size 5~25 mm, performance parameters see Table 2.

Recycled Aggregate: collection of waste concrete and reinforced concrete bulk containing impurities from construction waste; manual or mechanical removal of impurities such as large wood chunks; cutting or hammering pre-processing into concrete and reinforced concrete blocks; primary crushing using a rotor crusher; use of electromagnet sorting to remove iron and steel impurities; removal of impurities such as wood chunks and plastics; secondary crushing using a rotor crusher; rinsing to remove slurry; and use of a laboratory screening machine to screen the recycled coarse aggregate (particle size 5~25 mm) and recycled fine aggregate (particle size 0.15~5 mm). The performance parameters of recycled aggregate are shown in Table 2. Add the same amount of recycled fine aggregate, different amounts of recycled coarse aggregate, and natural coarse aggregate to the test block. 

Water: tap water.

### 3.2. Design of Concrete Mix Ratio

Since the C30 strength grade recycled concrete is more commonly used in the current concrete structural engineering, five recycled aggregate replacement rates for concrete were used for the design of recycled concrete proportion in this paper to compare the effect of freezing and thawing environment on the durability performance of recycled concrete with different recycled aggregate admixtures, as shown in Figure 1.

Based on the method of ordinary concrete proportion design (JGJ55-2011) [30], the proportion design of recycled concrete related to this test is carried out concerning the “ordinary concrete proportion design regulations” (JGJ55-2011). In the proportion design, the parameter water–cement ratio W/C is taken as 0.50, recycled fine aggregate rate is taken as 35%, and the detailed proportion design of recycled concrete is shown in Table 3.

### 3.3. Preparation of Recycled Aggregates

In this test, using the waste concrete blocks in the laboratory the waste concrete was selected to remove wood, steel, brick, plastic, and other garbage. The recycled concrete with impurities removed was initially crushed through the crusher to obtain the recycled aggregate, as shown in Figure 2. Then, the recycled aggregate was divided into recycled coarse aggregate (10–40 mm) and recycled fine aggregate (0–10 mm) using the screening machine. The recycled coarse aggregate and recycled fine aggregate were crushed twice and screened to obtain the recycled coarse aggregate (5–25 mm) and recycled fine aggregate (0.15–5 mm). The detailed process is shown in Figure 3. There was a secondary crushing of the recycled coarse aggregate and recycled fine aggregate, and after the screening, recycled coarse aggregate (5–25 mm) and recycled fine aggregate (0.15–5 mm) were obtained; the detailed process is shown in Figure 3. The quality of recycled aggregates is shown in Table 4.

### 3.4. Recycled Concrete Preparation

Recycled concrete is a composite material comprising cement paste, recycled coarse aggregate, recycled fine aggregate (or recycled fine aggregate), and other admixtures. The preparation of recycled concrete involves mixing a certain proportion of cement, water, and recycled aggregates in a mixer until thoroughly mixed to obtain fluid-recycled concrete. The freshly mixed recycled concrete is then placed into the membrane tool and cast together with the membrane tool on a concrete vibrating table, which is vibrated after compacting. After 24 h of indoor static, the membrane is removed, and the specimen is placed in a standard curing room with a temperature of 20 ± 3 °C, humidity of more than 95%. The detailed steps are illustrated in Figure 4. A total of 5 different recycled coarse aggregate content concrete specimens were prepared for the experiment, and different freeze–thaw cycles were conducted. The specimen design is shown in Figure 5.

### 3.5. Slump Test of Recycled Concrete

The workability of concrete is a critical factor in assessing the performance of concrete mixes for construction operations such as mixing, transportation, pouring, and pounding. It ensures consistent quality and dense compaction of concrete molding, influenced by three key features: cohesion and water retention. By conducting a slump test of recycled concrete with varying substitution rates (Figure 6), differences in fluidity, cohesion, and water retention were examined (shown in Table 4). This analysis has laid a theoretical groundwork for the practical application of recycled concrete in engineering projects.

### 3.6. Freeze–Thaw Cycles

To better understand how different recycled aggregates affect the mechanical properties and durability of concrete in cold climates, all specimens were subjected to standard curing conditions for 28 days before exposing them to 0, 15, and 30 freeze–thaw cycles. Each cycle involved exposing the concrete to −20 °C for 4 h followed by thawing at 20 °C for 20 h (Figure 7). The frost resistance test is conducted in accordance with the fast-freezing method in GB/T 50082-2009 [31].

This work aims to study the relationship between the mechanical property degradation and the number of freeze–thaw cycles and freeze–thaw damage, specifically about different admixtures of recycled aggregates. This is important because recycled concrete can experience significant degradation in cold regions, and we need to understand how to mitigate this damage.

### 3.7. Rebound Value Test

Rebound testing is a valuable nondestructive testing method; rebound testing can detect the strength of concrete. In this study, 45 recycled concrete cubic specimens were each tested at 16 test points, with 8 on the front and 8 on the back. The average rebound values of the 16 test points were calculated, with 3 maximum and 3 minimum values removed to obtain the average rebound value of each test block. The rebound value is calculated utilizing the following formula:$$Rm=\frac{\sum_{i=1}^{10}\mathrm{Ri}}{10}$$

Subsequently, the rebound nondestructive testing was repeated on 30 recycled concrete specimens that had completed 15 and 30 freeze–thaw cycles (Figure 8). This was performed to observe any changes in the rebound value of the recycled concrete specimens before and after freezing and thawing to determine the degree of damage to recycled concrete caused by freeze–thaw cycles.

### 3.8. Compressive Strength Test

As per the specifications outlined in the freeze–thaw cycle test design, it is necessary to halt the machinery after every 0, 15, and 30 freeze–thaw cycles. The cubic freeze–thaw specimen must then be extracted and left indoors for natural drying for 2 days before conducting the compressive test. The compressive strength test refers to the GBT50081-2002 [32] (Figure 9). The uniform loading method should be used during the compressive strength test with a loading speed of 0.05 MPa/s. 

### 3.9. Recycled Concrete Quality Loss Test

After being subjected to freeze–thaw cycles, the recycled concrete test block will undergo quality changes. Therefore, the cubic test blocks that have completed a set number of freeze–thaw cycles are wiped to remove the water indicated by the specimens and weighed using a high-precision electronic balance (with a sensitivity of 1 g) for the calculation of the mass loss rate of ΔW_n_. The average of the mass loss rate of three specimens is taken as the value of the ΔW_nm_ evaluation value after the freezing and thawing.

## 4. Results and Discussion

### 4.1. The Workability of Recycled Concrete

The usability of the recycled concrete samples underwent testing through the slump test and the findings are presented in Table 5. The results demonstrate a correlation between the compatibility and the admixture of recycled concrete, with the slump decreasing gradually as the percentage of recycled aggregate increases. For instance, the slump value of C30 concrete without recycled aggregate was 111 mm, while the slump of concrete with a 100% substitution rate was only 90 mm, indicating a notable difference of 21 mm. Compared to regular concrete, recycled concrete experiences a significant reduction in slump. Its fluidity performance is lower than that of ordinary concrete during the mixing process. The slump of recycled concrete is directly related to the substitution ratio of recycled aggregate, and it had a relationship with the substitution ratio of recycled aggregate. As the replacement ratio of recycled aggregate increases under the same water-cement ratio, the concrete slump gradually decreases. This is due to the porous nature and high water absorption of recycled aggregate, leading to decreased water content and slump. In addition, the rough surface of recycled aggregate increases mixing friction during the casting process, further reducing the slump.

### 4.2. Rebound Values

Rebound value is an important parameter for characterizing compressive strength, because measuring rebound value is lightweight, fast, has little impact on the structure, and has a fast detection speed. Construction units often use rebound value to check the development of concrete strength.

At a certain impact energy, the impact rod impacts the surface of the concrete, causing plastic deformation and consuming a portion of the work (the higher the strength and surface hardness of the concrete, the smaller the plastic deformation). Another portion of the work is transmitted back to the impact rod through the elastic deformation of the concrete, converting kinetic energy into elastic potential energy. The distance L1 at which the hammer rebounds backwards is the rebound value (Figure 10).

Figure 11 displays the outcomes of the standard procedure, indicating a decline in the rebound value of recycled concrete specimen decreases after freeze–thaw cycles. After 30 freeze–thaw cycles, the rebound value of concrete with a 50% replacement rate of recycled coarse aggregate decreased from 41 mm to 31 mm. The primary reason for this is due to the presence of uneven and irregular waste cement mortar in the recycled aggregate. The surface of cement mortar contains numerous pores and cracks, which can negatively impact its performance over time. As the number of freeze–thaw cycles increases, the freezing and expansion effect further contributes to the reduction in recycled aggregate performance, ultimately leading to a decrease in rebound values of the recycled concrete after freezing and thawing.

Compared with ordinary concrete, the rebound value of recycled concrete has decreased, and as the substitution rate of recycled coarse aggregate increases, the rebound value decreases more. The rebound value of ordinary concrete and concrete with 100% recycled coarse aggregate replacement rate decreased by 5 mm compared to each other. When the replacement rate of recycled aggregate does not exceed 50%, the rebound value decays slowly. When the replacement rate of recycled aggregate exceeds 50%, the rebound value decreases significantly (Figure 11). This effect is particularly pronounced in specimens that have a higher recycled aggregate ratio, as the lower strength of the recycled concrete aggregate contributes to a decrease in overall concrete strength. Notably, as the proportion of recycled concrete aggregate admixture increases, the decrease in concrete strength becomes more apparent. 

For different measuring points on the same test block, the rebound value of concrete varies, and sometimes the rebound value may suddenly increase. The reason for this phenomenon (Figure 12) is the uneven distribution of recycled coarse aggregates in concrete and the varying hardness of recycled aggregates.

### 4.3. Changes in Quality Loss

Figure 13 depicts the correlation between the loss rate of recycled concrete, the number of freeze–thaw cycles, and the replacement rate of recycled aggregate after 15 and 30 cycles, respectively. After 15 cycles of freeze–thaw cycles, there was a slight loss in quality between the freeze–thaw and unfrozen recycled concrete, but it was not significant. After 30 cycles of freeze–thaw cycles, there was a significant loss in quality between the freeze–thaw and unfrozen recycled concrete. Moreover, after the same freeze–thaw cycle of 30 times, the concrete (R100) completely using recycled aggregates showed more quality loss compared to the concrete (RC) using natural aggregates. From this, the following conclusion can be drawn: 

As the replacement rate of recycled aggregate increases, the quality loss of recycled concrete after freeze–thaw cycles gradually increases. Additionally, the number of freeze–thaw cycles significantly impacts the quality loss of recycled concrete, with more drastic changes observed as the number of cycles increases.

A key factor contributing to this phenomenon is an uneven layer of irregular waste mortar on the surface of recycled aggregate. This layer creates an interfacial transition zone with the newly mixed cement mortar, which is not sufficiently hydrated and results in an unstable bond. During the freeze–thawing cycle, the discontinuity and instability of the mortar located on the surface make it susceptible to detachment. Due to the freezing and swelling effect, this phenomenon becomes more prominent.

### 4.4. Compressive Strength

Overall, the compressive strength of recycled concrete decreases with increasing freeze–thaw cycles. The compressive strength of test blocks with a replacement rate of recycled coarse aggregate less than 75% decreases slowly, while when it exceeds 75%, the compressive strength decreases significantly (Figure 14).

When subjected to 15 freeze–thaw cycles, the substitution rate of recycled aggregates had little effect on compressive strength. Compared with R100%, R0% reduced the compressive strength loss rate by 4.25%. After 30 freeze–thaw cycles, the substitution rate of recycled aggregates has a significant effect on compressive strength. Compared with R100%, R0% reduced compressive strength by 15.66% (Figure 15).

The decrease in compressive strength is related to the material properties of recycled aggregates. The irregular size and shape of cement particles prevent seamlessly connected with the aggregate resulting in voids near the aggregates that are filled with small particles and molecules. This phenomenon, known as the “wall effect”, is caused by a layer of spherical cover around the aggregate called the interfacial transition zone (ITZ) [33]. Due to the abundance of small particles and voids within the ITZ, the bearing capacity of this zone is lower than that of ordinary mortar. Recycled concrete features two interfacial transition zones. The first, ITZ2, surrounds the natural aggregate, while the second, ITZ1, forms around the old mortar during casting. These zones create small voids, contributing to recycled concrete’s higher porosity than ordinary concrete. This porosity also contributes to its lower load-bearing capacity. And the increased porosity also weakens its mechanical properties and makes it more susceptible to erosion of water, chloride ions, and CO_2_. Additionally, during the preparation process, irreversible cracks can form in the old mortar layer, making it susceptible to crack expansion during the loading and ultimately leading to the failure of the recycled concrete.

According to research by Park and Noguchi [34], impurities such as wood, plastic, and aluminum found in recycled aggregate can impact the mechanical properties of recycled concrete. For instance, if the aggregate contains 0.05% aluminum impurities (size 2.5–5 mm), it can reduce the compressive strength of recycled aggregate concrete by up to 7%. Additionally, the quality of recycled concrete can be affected by the production process of recycled aggregate. As the number of times recycled aggregate is crushed increases, the density of recycled aggregate increases and water absorption decreases, which can result in lower strength of recycled concrete.

## 5. Enhanced Performance of Recycled Concrete

We found from the experiment that with the increase of recycled aggregate content and freeze–thaw cycles, the performance of concrete decreases. The following summarizes some methods to improve the performance of recycled concrete. According to research by Guo et al. [35], incorporating volcanic ash material as a coating or blending it into recycled concrete can significantly enhance its durability. Similarly, Katz et al. [36] discovered that impregnating recycled aggregate in a silica fume solution and ultrasonic cleaning can improve the properties of recycled concrete. Tam et al. [37] proposed a novel mixing method called the two-stage mixing method, which involves adding water during the first stage of mixing to create a thin layer of cement paste on the surface of the recycled aggregate. This paste fills in micro-cracks of the old cement mortar, enhancing the strength and integrity of the material. During the second stage of mixing, the remaining water was added to complete the concrete mixing process, and the compressive strength of the recycled concrete was increased by 21.90% through the two-stage mixing method. Choi et al. [38] proposed a surface modification technique to enhance the properties of recycled concrete. This technique involves applying a coarse slurry comprising of inorganic admixtures to the surface of recycled aggregate. The results of this treatment showed an increase of 15% in compressive strength and 30% in shear strength.

One distinguishing factor between recycled and natural concrete is the cement mortar attached to the recycled aggregate [39]. In order to enhance the performance of recycled concrete, it is essential to minimize the amount of cement mortar that remains on the recycled aggregate. Koshiro and Ichise [40] have explored various techniques, such as heating and grinding to effectively remove the cement mortar and produce high-quality recycled aggregate from concrete waste. Kim et al. [41] improved the quality of recycled aggregate by crushing and grinding it in a sulfuric acid solution. In addition, there are various methods to improve the quality of recycled aggregates, such as electrical pulse power [42], acid milling [43], carbonation of recycled aggregates [44], and microorganisms [45].

## 6. Conclusions

As the replacement rate of recycled aggregate increases, the workability of recycled concrete is negatively impacted. The slump of concrete mixed with 100% recycled coarse aggregate decreased by 21 mm compared to concrete without recycled coarse aggregate. This is primarily attributed to numerous pore cracks on the surface of recycled aggregate, along with its high water absorption, subpar surface roughness, and limited fluidity.The rebound value of recycled concrete tends to decrease as the rate of recycled substitution increases. The maximum attenuation of rebound value is 8 mm. After 30 freeze–thaw cycles, the rebound value of concrete mixed with 50% recycled coarse aggregate decreased by 8 mm. When the replacement rate of recycled aggregate is less than 50%, the loss of rebound value is not severe, and the performance of recycled concrete is good. Therefore, when using recycled concrete in cold regions, the amount of waste mortar on recycled aggregates should be minimized as much as possible. When the recycled coarse aggregate does not exceed 50%, the concrete exhibits good performance.The higher the number of freeze–thaw cycle, the higher the rate of quality loss. However, the overall quality loss is not severe, with only a slight detachment of the surface concrete.When the replacement rate of recycled coarse aggregate exceeds 75%, the compressive strength of the concrete decreases significantly. In practical applications, it is not advisable to use too much recycled aggregate. The influence of the replacement rate of recycled aggregates on strength becomes sensitive as the degree of freeze–thaw cycles increases. Therefore, when using recycled concrete in particularly cold places, the substitution rate of recycled coarse aggregates should be reduced to achieve good mechanical properties.The strength and lifespan of recycled concrete can be enhanced through various methods such as heat treatment, chemical treatment, or the addition of organic materials or other solvents to the recycled aggregate.

This research serves as a guide for creating and using recycled concrete in cold regions.

## Figures and Tables

**Figure 1 materials-17-03934-f001:**
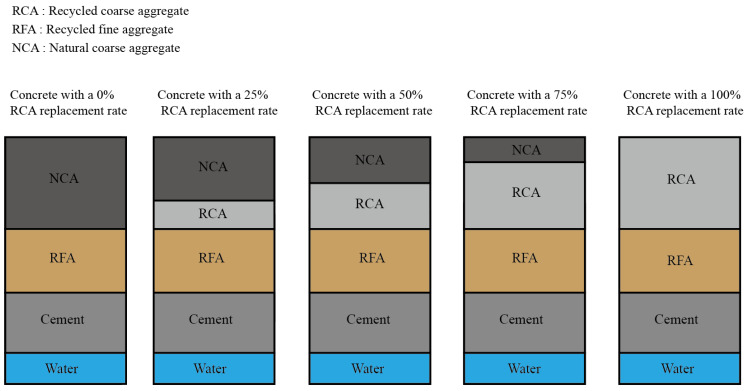
Design of recycled concrete mix ratio.

**Figure 2 materials-17-03934-f002:**
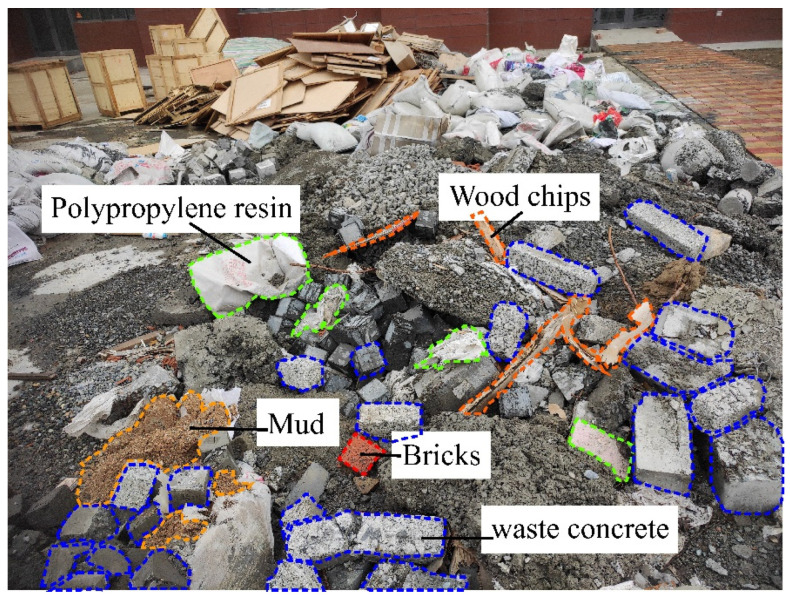
Sources of recycled aggregates.

**Figure 3 materials-17-03934-f003:**
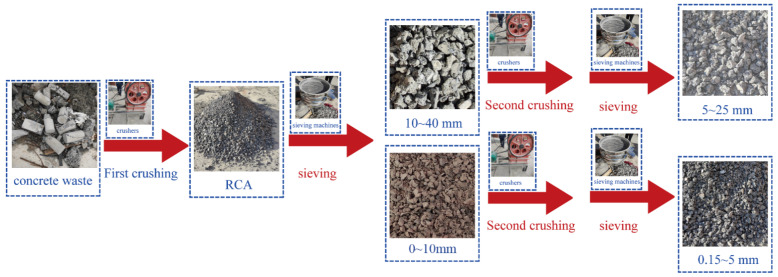
Preparation process of recycled aggregate.

**Figure 4 materials-17-03934-f004:**
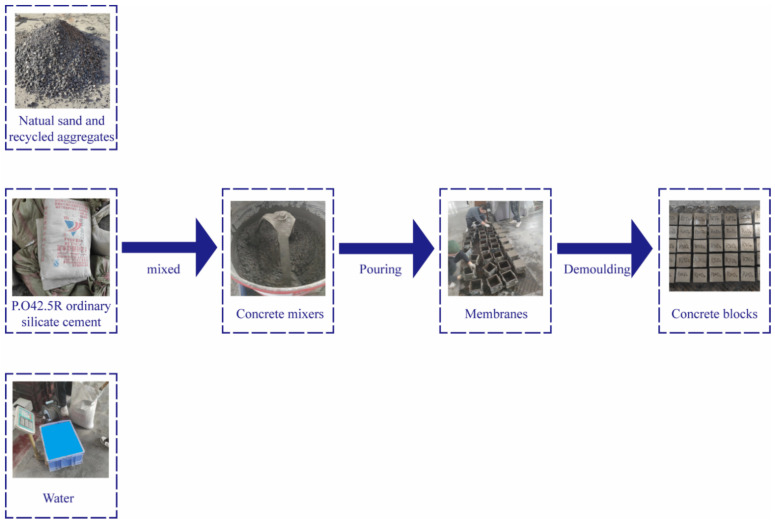
Flow chart of production of recycled concrete.

**Figure 5 materials-17-03934-f005:**
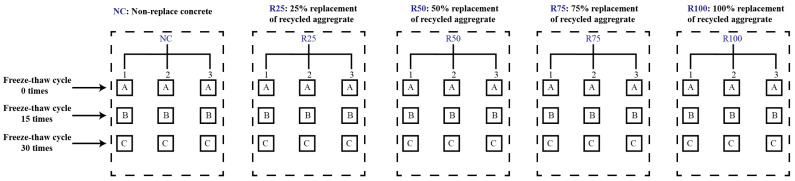
Classification chart of all concrete test blocks.

**Figure 6 materials-17-03934-f006:**
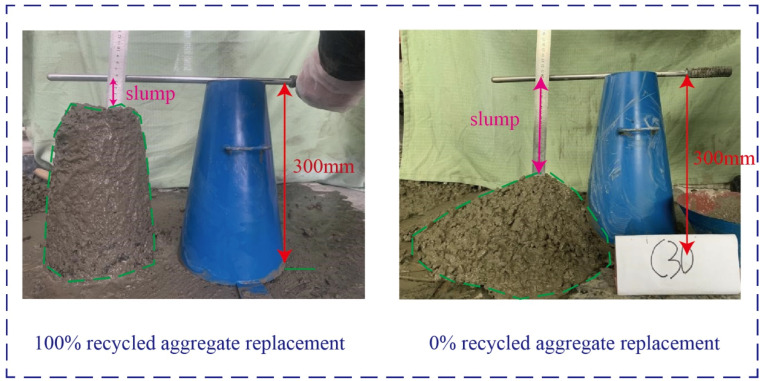
Slump test with different recycled aggregate substitution rates.

**Figure 7 materials-17-03934-f007:**
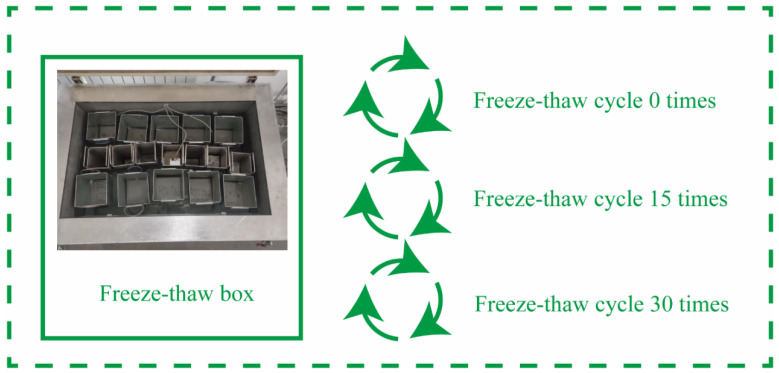
Freeze–thaw cycle test.

**Figure 8 materials-17-03934-f008:**
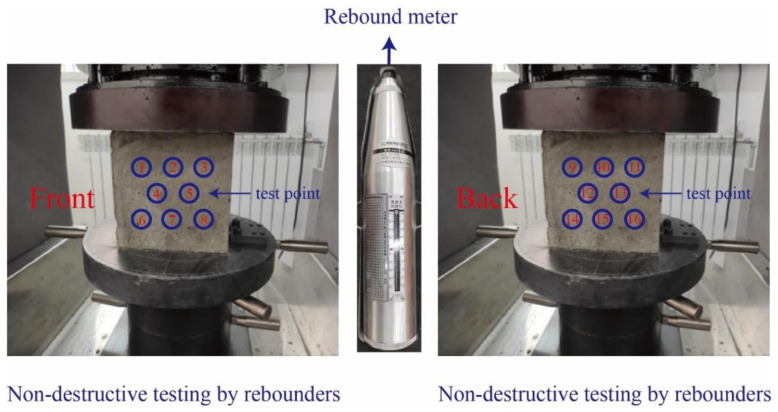
Rebound value test of recycled concrete.

**Figure 9 materials-17-03934-f009:**
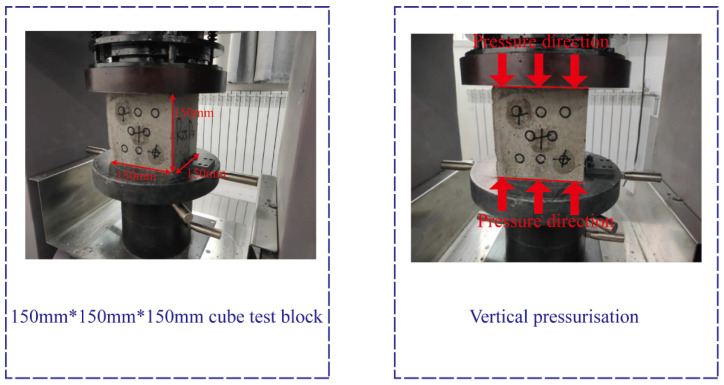
Compressive strength test.

**Figure 10 materials-17-03934-f010:**
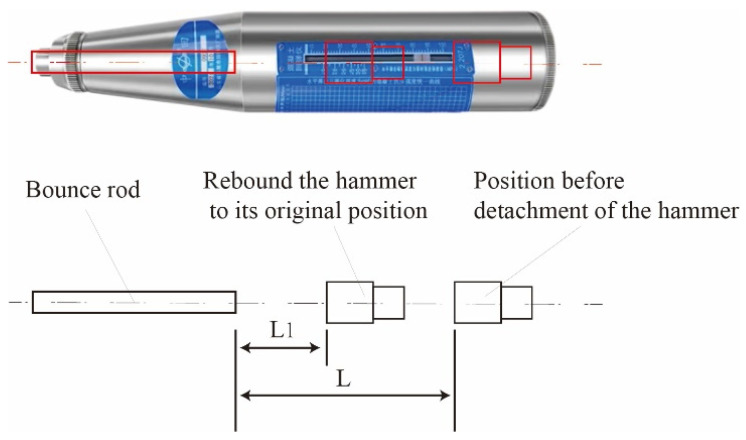
Rebound value explanation diagram.

**Figure 11 materials-17-03934-f011:**
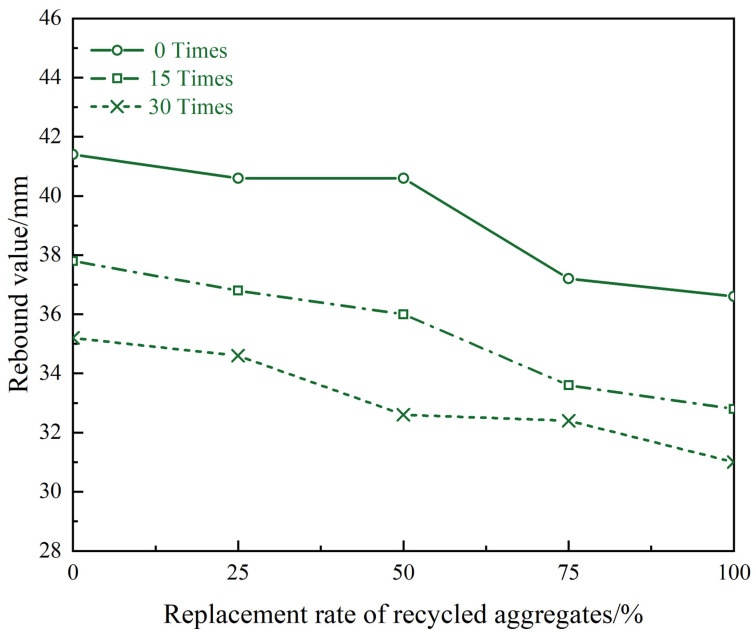
Rebound values of recycled concrete with different substitution rates under different numbers of freeze–thaw cycles.

**Figure 12 materials-17-03934-f012:**
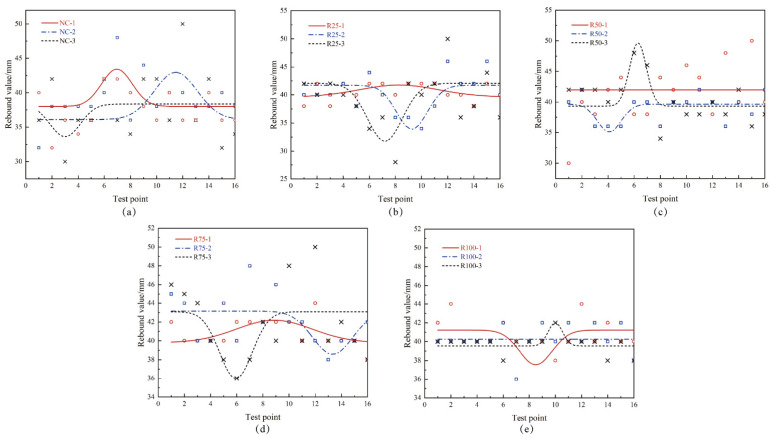
Rebound values of recycled concrete with 0% (**a**), 25% (**b**), 50% (**c**), 75% (**d**), 100% (**e**) recycled aggregate rate.

**Figure 13 materials-17-03934-f013:**
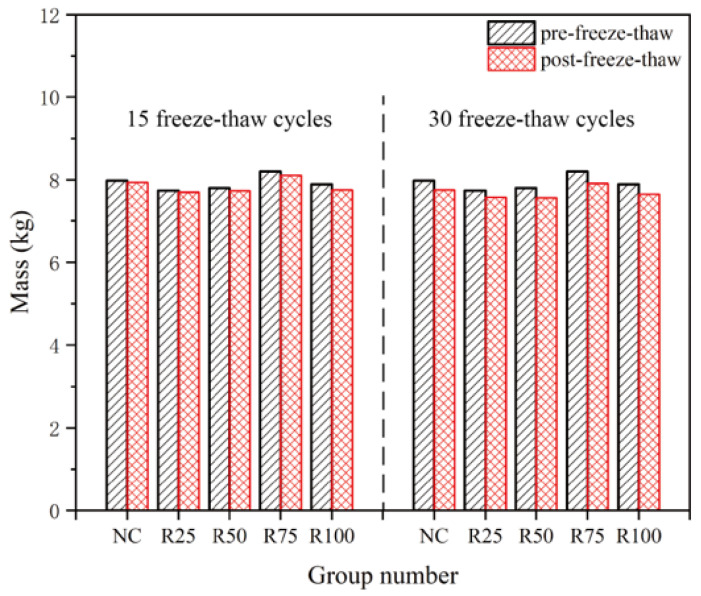
Changes in mass loss.

**Figure 14 materials-17-03934-f014:**
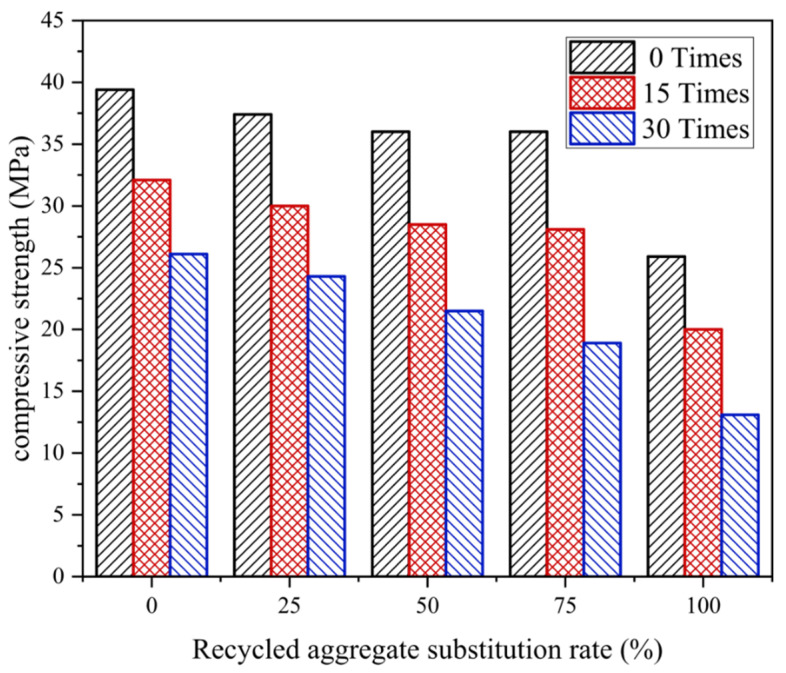
Change in compressive strength.

**Figure 15 materials-17-03934-f015:**
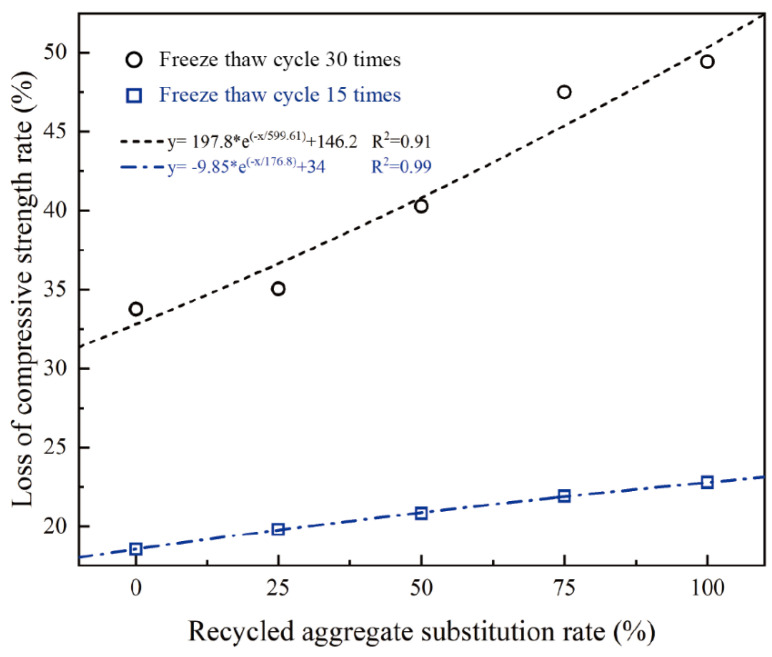
Compressive strength loss rate.

**Table 1 materials-17-03934-t001:** Main chemical compositions of cement %.

CaO	SiO_2_	Al_2_O_3_	Fe_2_O_3_	MgO	SO_3_
57.65	21.44	5.36	3.56	1.24	2.43

**Table 2 materials-17-03934-t002:** Performance parameters of aggregates.

Categories	Apparent Density (kg/m^3^)	Packing Density (kg/m^3^)	Indicators of Crushing (%)	24 h Water Absorption (%)
Natural coarse aggregate	2738	1541	7.3	0.97
Recycled fine aggregate	2840	1443	6	2.76
Recycled coarse aggregate	2514	1427	14.2	6.14

**Table 3 materials-17-03934-t003:** Recycled concrete mix ratio design.

Specimen Type	Water-to-Cement Ratio	Amount of Material in Recycled Concrete (kg/m^−3^)					
		Cement	Water	Recycled Fine Aggregate	Coarse Aggregate		
					Natural	Renewable	Replacement Rate (%)
NC	0 50	411.5	205.8	603.6	1207.1	0	0
R25	0.50	411.5	205.8	603.6	905.3	301.8	25
R50	0.50	411.5	205.8	603.6	603.5	603.5	50
R75	0.50	411.5	205.8	603.6	301.8	905.3	75
R100	0.50	411.5	210	610.8	0	1207.1	100

**Table 4 materials-17-03934-t004:** Quality of recycled aggregates.

Technical Index	Content (Mass)/%
Micro powder content	1.51
Clay lump	0.64
Water absorption rate	4.24
Needle-like particle content	5
Sulfide and sulfate content	1.11
Chloride content	0.02
Impurity content	0.6
Mass loss	6.57
Crushing index	15
Apparent density/(kg/m^3^)	2391
Void ratio	47

**Table 5 materials-17-03934-t005:** The workability of recycled concrete.

Concrete Type	Compatibility		
	Slump	Cohesive	Water Retention
NC	111	A	A
R25	101	B	B
R50	98	A	B
R75	96	A	A
R100	90	A	B

Note: A is good, B is fair.

## Data Availability

The original contributions presented in the study are included in the article, further inquiries can be directed to the corresponding author.

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
