# Peer review of "Experimental Study of Recycled Concrete under Freeze–Thaw Conditions"

_materials, 2024, doi:10.3390/ma17163934_

Round 1

Reviewer 1 Report

Comments and Suggestions for Authors

Dear authors,

In this manuscript, Experimental study of recycled concrete under freeze-thaw conditions by authors Alipujiang Jerula, Cong Wu, Zhixuan Fu, Hushitaer Niyazi,Haodong Li, investigated possibility of applying recycled concrete to cold and severe regions. Regarding that the authors pointed necessity to consider the freeze thaw resistance of recycled concrete. Using an indoor freeze-thaw cycle test system, the rapid freezing method was used to conduct rapid freeze-thaw tests on recycled concrete specimens under set freeze-thaw cycle conditions. The influence of various factors on the compressive strength, quality loss rate, and rebound value of recycled concrete was analyzed. The results indicate that with the increase of freeze-thaw cycles and recycled aggregate content, the workability, rebound value, and compressive strength of the specimens decrease, and the quality loss rate increases. The overall quality loss rate shows a decreasing trend, but the quality loss is not severe. The authors concluded this research serves as a guide for creating and using recycled concrete in cold regions.

The manuscript is interesting and offers the possibility of further research in this area, especially as far as quality and quantity of recycled concrete. It is very important to improve the quality of recycled concrete at the chemical level, regardless of the purposes for which it will be used further. In their further research, the authors could focus more on structural characterization using certain instrumental and microscopic methods.

I have a few remarks that I would like the authors to answer or to correct the manuscript in that sense.

1.      It is necessary to follow the instructions for authors, when it comes to citing references in the manuscript.

2.      Although you devoted one whole chapter (3.2.) to results of rebound values of recycled concrete, you have to explain the concept and importance of rebound value.

3.      Lines 180-182 -"The frost resistance test is conducted in accordance with the fast 180 freezing method in GB/T 50082-2009 Standard of Test Methods for Long-term Performance and Durability of Ordinary Concrete."

Reference/s after this sentence is/are required.

4.      Line 209-210- "This test should be performed using the relevant compressive strength test methods in the literature."

What literature did you use? Please provide the reference/s.

5.      Lines 263- Before Fig 12 (not after), it is necessary to explain what is in Fig 12.

6.      This previous comment (5) applies to Figures 13 and 14 and, of course, to every Figs and Table in the text of the manuscript.

Sincerely

Author Response

Comments 1: It is necessary to follow the instructions for authors, when it comes to citing references in the manuscript.

Response 1: We fully agree with your viewpoint and have checked and made revisions in the paper.

Comments 2: Although you devoted one whole chapter (3.2.) to results of rebound values of recycled concrete, you have to explain the concept and importance of rebound value.

Response 2: Thank you for the reviewer's comments. We should indeed explain the concept and importance of rebound value. We have added this section (See Chapter 3.2 of the revised manuscript)

Figure 10 Rebound Value Explanation Diagram

Rebound value is an important parameter for characterizing compressive strength, because measuring rebound value is lightweight, fast, has little impact on the structure, and has a fast detection speed. Construction units often use rebound value to check the development of concrete strength.

At a certain impact energy, the impact rod impacts the surface of the concrete, causing plastic deformation and consuming a portion of the work (the higher the strength and surface hardness of the concrete, the smaller the plastic deformation). Another portion of the work is transmitted back to the impact rod through the elastic deformation of the concrete, converting kinetic energy into elastic potential energy. The distance L1 at which the hammer rebounds backwards is the rebound value. (Fig-ure 10) (Section 3.2, line 253)

Comments 3: Lines 180-182 -"The frost resistance Long-term Performance and Durability of Ordinary Concrete test is conducted in accordance with the fast 180 freezing method in GB/T 50082-2009 Standard of Test Methods for." Reference/s after this sentence is/are required.

Response 3: Thank you very much for your feedback. We have added the corresponding reference after this sentence.

Chinese National Standard, Standard for test methods of long term performance and durability of ordinary concrete, GB/T 50082-2009. Beijing, China, 2009. (Reference, line 452)

Comments 4: Line 209-210- "This test should be performed using the relevant compressive strength test methods in the literature. "What literature did you use? Please provide the reference/s.

Response 4: Thank you very much for your valuable feedback. We have added specific literature references for the experiment.

Compressive strength test reference to Chinese National Standard, Standard for test method of mechanical properties on ordinary concrete, GBT50081–2002.[18] (Section 2.8, line 219)

Chinese National Standard, Standard for test method of mechanical properties on ordinary concrete, GBT50081–2002. Beijing, China, 2003. (Reference, line 454)

Comments 5: Lines 263- Before Fig 12 (not after), it is necessary to explain what is in Fig 12.

Response 5: Thank you for pointing out the issue to us. We have changed the position of Figure 12 and explained its content

After 15 cycles of freeze-thaw cycles, there was a slight loss in quality between the freeze-thaw and unfrozen recycled concrete, but it was not significant. After 30 cycles of freeze-thaw cycles, there was a significant loss in quality between the freeze-thaw and unfrozen recycled concrete. Moreover, after the same freeze-thaw cycle of 30 times, the concrete (R100) completely using recycled aggregates showed more quality loss compared to the concrete (RC) using natural aggregates. (Section 3.3, line 291)

Comments 6: This previous comment (5) applies to Figures 13 and 14 and, of course, to every Figs and Table in the text of the manuscript.

Response 6: Thank you for pointing out the issue to us. We have carefully checked the analysis section corresponding to each graph and supplemented the missing parts.

Overall, the compressive strength of recycled concrete decreases with increasing freeze-thaw cycles. The compressive strength of test blocks with a replacement rate of recycled coarse aggregate less than 75% decreases slowly, while when it exceeds 75%, the compressive strength decreases significantly. (Figure 14) (Section 3.4, line 316)

When subjected to 15 freeze-thaw cycles, the substitution rate of recycled aggre-gates had little effect on compressive strength. Compared with R100%, R0% reduced the compressive strength loss rate by 4.25%. After 30 freeze-thaw cycles, the substitu-tion rate of recycled aggregates has a significant effect on compressive strength. Com-pared with R100%, R0% reduced compressive strength by 15.66%. (Figure 15) (Section 3.4, line 320)

Reviewer 2 Report

Comments and Suggestions for Authors

The manuscript covers an interesting topic. Nevertheless, it requires corrections.

In the Introduction section, the specific usefulness of the research carried out and its novelty should be indicated.

The initial literature review is poor and needs to be expanded.

Please indicate, based on the analysis, which mixture composition gave the most optimal results and which may be desirable for wider use and why.

The text requires deep editing improvement. Some errors are unacceptable.

Author Response

Comments 1:In the Introduction section, the specific usefulness of the research carried out and its novelty should be indicated.

Response 1: Thank you for pointing out the issue for us. We have added the novelty and usefulness of this study in the introduction section.

At present, there is no research on the frost resistance of recycled concrete under various replacement rates of recycled coarse aggregates.

It is worth mentioning that when using recycled concrete in cold regions, attention should be paid to minimizing the amount of waste mortar on the surface of recycled aggregates. The replacement rate of recycled coarse aggregates can be selected ac-cording to actual strength needs, but good performance can be achieved when it does not exceed 50%. (Section 1, line 105)

Comments 2: The initial literature review is poor and needs to be expanded.

Response 2: Thank you for your suggestion. We have supplemented the literature review section.

When concrete reach their service life, how to dispose of the waste concrete is a big problem. These piles of construction wastes can consume land resources and pollute water and soil. There are many sources of waste concrete, such as aging, explosions, earthquakes, floods, and typhoons. It is very necessary to reuse the solid waste from construction. Solid waste concrete has significant environmental, economic, and sus-tainable benefits. [2] Recycled aggregates produced from solid waste also have good mechanical properties, which can meet the demand for aggregates in the construction industry. [3] (Section 1, line 37)

Comments 3: Please indicate, based on the analysis, which mixture composition gave the most optimal results and which may be desirable for wider use and why.

Response 3: Thank you for pointing out the issue. We have carefully revised this section.

The replacement rate of recycled aggregate is less than 50%, the loss of rebound value is not severe, and the performance of recycled concrete is good. Therefore, when using recycled concrete in cold regions, the amount of waste mortar on re-cycled aggregates should be minimized as much as possible. When the recycled coarse aggregate does not exceed 50%, the concrete exhibits good performance. (Section 5, line 385)

When the replacement rate of recycled coarse aggregate exceeds 75%, the com-pressive strength of concrete decreases significantly. In practical applications, it is not advisable to use too much recycled aggregate. The influence of the replace-ment rate of recycled aggregates on strength becomes sensitive as the degree of freeze-thaw cycles increases. Therefore, when using recycled concrete in particu-larly cold places, the substitution rate of recycled coarse aggregates should be re-duced to achieve good mechanical properties. (Section 5, line 394)

Comments 4: The text requires deep editing improvement. Some errors are unacceptable.

Response 4: Thank you for pointing out the issue. We will carefully examine and make necessary modifications.

Reviewer 3 Report

Comments and Suggestions for Authors

The paper aims to investigate the mechanical properties of recycled concrete by studying the number of freezing and thawing cycles and the rate of recycled coarse aggregate substitution. The experiment adopts the fast freezing method to conduct rapid freeze-thaw tests on specimens with different replacement rates of recycled coarse aggregates.

The article is clearly written and results are clearly presented. The reviewer suggests for its publication after the following minor comments are addressed:

·       The abstract is quite long. Following the MPDI suggestions, abstracts should not exceed 250 words. Authors are suggested to reduce the abstract, especially the last part where too details are added for an abstract.

·       In the first part of the introduction authors addressed the topic of waste concrete and its shortcomings. However, more literature should be added, better explaining which are the causes of waste concrete (i.e., aging, exceptional events, such as earthquakes, floods, hurricanes, etc.). To support this task, the reviewer suggests some article that could be added by the authors:

o   Yang Y. et al. A comprehensive review of multisource solid wastes in sustainable concrete: From material properties to engineering application. Construction and building materials, Vol. 435, 136775, 2024.

o   Chandru U. et al. Systematic comparison of different recycled fine aggregates from construction and demolition wastes in OPC concrete and PPC concrete. Journal of Building Engineering, Vol. 75, 106768, 2023.

o   Nicoletti V., Carbonari S., Gara F. Nomograms for the pre-dimensioning of RC beam-column joints according to Eurocode 8. Structures, 39, 958-973, 2022. DOI: 10.1016/j.istruc.2022.03.083.

·       Section 2.1 should be better introduced. Are those materials relevant to the specimen prepared by the authors for their experimentation?

·       Text in Fig. 2 is not very readable. Please, improve the figure quality. The same for Fig. 5 (the text here is too small).

·       Section 3.5 is very important since suggestions are provided to improve the concrete quality. Authors are suggested to dedicate a new section (i.e., section 4) for this part.

·       Conclusions must be better explained. So far, they summarize the main findings of the research. However, the research motivations and the main achievements should be clearly stated, as well as the utilities (and possible practical applications) of the obtained results.

·       References must be placed within square brackets in the manuscript text.

Author Response

Comments 1: The abstract is quite long. Following the MPDI suggestions, abstracts should not exceed 250 words. Authors are suggested to reduce the abstract, especially the last part where too details are added for an abstract.

Response 1: Thank you for your valuable feedback. We have made revisions to the abstract section.

Comments 2: In the first part of the introduction authors addressed the topic of waste concrete and its shortcomings. However, more literature should be added, better explaining which are the causes of waste concrete (i.e., aging, exceptional events, such as earthquakes, floods, hurricanes, etc.). To support this task, the reviewer suggests some article that could be added by the authors:

o   Yang Y. et al. A comprehensive review of multisource solid wastes in sustainable concrete: From material properties to engineering application. Construction and building materials, Vol. 435, 136775, 2024.

o   Chandru U. et al. Systematic comparison of different recycled fine aggregates from construction and demolition wastes in OPC concrete and PPC concrete. Journal of Building Engineering, Vol. 75, 106768, 2023.

o   Nicoletti V., Carbonari S., Gara F. Nomograms for the pre-dimensioning of RC beam-column joints according to Eurocode 8. Structures, 39, 958-973, 2022. DOI: 10.1016/j.istruc.2022.03.083.

Response 2: Thank you for your valuable suggestion. We have supplemented the introduction section (reasons for concrete waste).

When concrete reach their service life, how to dispose of the waste concrete is a big problem. These piles of construction wastes can consume land resources and pollute water and soil. There are many sources of waste concrete, such as aging, explosions, earthquakes, floods, and typhoons. It is very necessary to reuse the solid waste from construction. Solid waste concrete has significant environmental, economic, and sustainable benefits. 2 Recycled aggregates produced from solid waste also have good mechanical properties, which can meet the demand for aggregates in the construction industry. 3(Section 1, line 37)

Comments 3: Section 2.1 should be better introduced. Are those materials relevant to the specimen prepared by the authors for their experimentation?

Response 3: Thank you for your valuable feedback. We have added it to section 2.1.

Add the same amount of recycled fine aggregate, different amounts of recycled coarse aggregate, and natural coarse aggregate to the test block.(Section 2.1, line 125)

Comments 4: Text in Fig. 2 is not very readable. Please, improve the figure quality. The same for Fig. 5 (the text here is too small).

Response 4: Thank you for pointing out the issue to us. We have replaced the image in Figure 2.

Figure 2. Sources of recycled aggregates. (Section 2.3, line 156)

Comments 5: Section 3.5 is very important since suggestions are provided to improve the concrete quality. Authors are suggested to dedicate a new section (i.e., section 4) for this part.

Response 5: Thank you for your valuable feedback. We have revised section 3.5 to 4

  1. Enhanced Performance of Recycled ConcreteSection 4, line 342)

Comments 6: Conclusions must be better explained. So far, they summarize the main findings of the research. However, the research motivations and the main achievements should be clearly stated, as well as the utilities (and possible practical applications) of the obtained results.

Response 6: Thank you for your valuable feedback. We have supplemented the conclusion section with our opinions.

The replacement rate of recycled aggregate is less than 50%, the loss of rebound value is not severe, and the performance of recycled concrete is good. Therefore, when using recycled concrete in cold regions, the amount of waste mortar on re-cycled aggregates should be minimized as much as possible. When the recycled coarse aggregate does not exceed 50%, the concrete exhibits good performance. (Section 5, line 386)

When the replacement rate of recycled coarse aggregate exceeds 75%, the com-pressive strength of concrete decreases significantly. In practical applications, it is not advisable to use too much recycled aggregate. The influence of the replace-ment rate of recycled aggregates on strength becomes sensitive as the degree of freeze-thaw cycles increases. Therefore, when using recycled concrete in particu-larly cold places, the substitution rate of recycled coarse aggregates should be re-duced to achieve good mechanical properties. (Section 5, line 395)

Comments 7: References must be placed within square brackets in the manuscript text.

Response 7: Thank you for pointing out the issue. We have added square brackets.

Reviewer 4 Report

Comments and Suggestions for Authors

1.       Lacks Detailed Methodological Justification:

-        The article does not provide a thorough justification for the chosen experimental methods; particularly why specific freeze-thaw cycle parameters and replacement rates were selected. This omission weakens the rationale behind the experimental design.

-        Reference: "Experimental Program" (p. 3, heading 2), "Freeze-thaw cycles" (p. 7, heading 2.5).

2.       Insufficient Data on Aggregate Quality:

-        The study inadequately details the initial quality of recycled aggregates, such as variations in impurities and mechanical properties before use. This lack of information makes it difficult to assess the consistency and reliability of the experimental results.

-        Reference: "Experimental Program" (p. 3, heading 2), "Preparation of recycled aggregates" (p. 5, heading 2.4).

3.       Insufficient Data on Aggregate Quality:

-        The article does not discuss the extended durability and performance of recycled concrete beyond the 30 freeze-thaw cycles. This constraint hinders a thorough comprehension of the material's performance over prolonged durations in actual environmental circumstances.

-        Reference: "Experimental Program" (p. 3, heading 2), "Preparation of recycled aggregates" (p. 4, heading 2.3)

4.       Neglects Environmental Impact Assessment:

-        The research does not evaluate the environmental impact of using recycled concrete, such as potential benefits in reducing carbon footprint or waste. Including such an assessment would strengthen the argument for its sustainability.

-        Reference: "Introduction" (p. 1, heading 1)

5.       Inadequate Comparative Analysis:

-        The article does not provide a robust comparative analysis between recycled concrete and conventional concrete in various environmental conditions. More detailed comparisons would help in better understanding the advantages and disadvantages of recycled concrete.

-        Reference: "Results and discussion" (p. 10, heading 5)

Comments on the Quality of English Language

Moderate

Author Response

Thank you for your letter and for the reviewers’ comments concerning our manuscript entitled “Experimental study of recycled concrete under freeze-thaw conditions” (ID: materials-3109200).Those comments are all valuable and very helpful for revising and improving our paper, as well as the important guiding significance to our researches. We have studied comments carefully and have made correction which we hope meet with approval.

Revised portion are marked in red in the revised manuscript. The main corrections in the paper and the responds to the reviewer’s comments are as flowing:

1.Lacks Detailed Methodological Justification:

The article does not provide a thorough justification for the chosen experimental methods; particularly why specific freeze-thaw cycle parameters and replacement rates were selected. This omission weakens the rationale behind the experimental design.

Reference: "Experimental Program" (p. 3, heading 2), "Freeze-thaw cycles" (p. 7, heading 2.5).

Thank you for your valuable feedback. We have provided additional explanations on why the current experimental protocol was chosen

In order to investigate the effect of recycled coarse aggregate content on the frost re-sistance of concrete, recycled coarse aggregates with replacement rates of 0%, 25%, 50%, 75%, and 100% were selected. This is beneficial for us to comprehensively observe the influence of replacement rates on the frost resistance of concrete. Secondly, due to the significantly lower mechanical properties of recycled concrete compared to ordi-nary concrete and the severe damage caused by excessive freeze-thaw cycles, it is not conducive to analyzing the impact of recycled aggregates on concrete. Therefore, 15 freeze-thaw cycles and 30 freeze-thaw cycles were selected.(Page 6,Section 2, line 175)

2.Insufficient Data on Aggregate Quality:

The study inadequately details the initial quality of recycled aggregates, such as variations in impurities and mechanical properties before use. This lack of information makes it difficult to assess the consistency and reliability of the experimental results.

Reference: "Experimental Program" (p. 3, heading 2), "Preparation of recycled aggregates" (p. 5, heading 2.4).

Thank you for your valuable feedback. We have added the quality of recycled aggregates in section 3.3.

The quality of recycled aggregates is shown in Table 4. (Page 6, Section 3.3, Line 231)

Table 4 Quality of recycled aggregates

Technical Index

Content (mass)/%

Micro powder content

1.51

clay lump

0.64

Water absorption rate

4.24

Needle like particle content

5

Sulfide and sulfate content

1.11

Chloride content

0.02

impurity content

0.6

Mass loss

6.57

crushing index

15

Apparent density/(kg/m3)

2391

Void ratio

47

3.Insufficient Data on Aggregate Quality:

The article does not discuss the extended durability and performance of recycled concrete beyond the 30 freeze-thaw cycles. This constraint hinders a thorough comprehension of the material's performance over prolonged durations in actual environmental circumstances.

Reference: "Experimental Program" (p. 3, heading 2), "Preparation of recycled aggregates" (p. 4, heading 2.3)

Thank you for your valuable feedback. We should indeed clarify this section and have made revisions and additions to it.

After 30 freeze-thaw cycles, the rebound value of concrete with a 50% replacement rate of recycled coarse aggregate decreased from 41mm to 31mm. the primary reason for this is due to the presence of uneven and irregular waste cement mortar in the re-cycled aggregate. The surface of cement mortar contains numerous pores and cracks, which can negatively impact its performance over time. As the number of freeze-thaw cycles increases, the freezing and expansion effect further contributes to the reduction in recycled aggregate performance, ultimately leading to a decrease in rebound values of the recycled concrete after freezing and thawing. (Page 12, Section 4.2, Line 345)

After 15 cycles of freeze-thaw cycles, there was a slight loss in quality between the freeze-thaw and unfrozen recycled concrete, but it was not significant. After 30 cycles of freeze-thaw cycles, there was a significant loss in quality between the freeze-thaw and unfrozen recycled concrete. Moreover, after the same freeze-thaw cycle of 30 times, the concrete (R100) completely using recycled aggregates showed more quality loss compared to the concrete (RC) using natural aggregates. From this, the following conclusion can be drawn:

As the replacement rate of recycled aggregate increases, the quality loss of recy-cled concrete after freeze-thaw cycles gradually increases. Additionally, the number of freeze-thaw cycles, significantly impact the quality loss of recycled concrete, with more drastic changes observed as the number of cycles increases. (Page 14, Section 4.3, Line 374)

When subjected to 15 freeze-thaw cycles, the substitution rate of recycled aggregates had little effect on compressive strength. Compared with R100%, R0% reduced the compressive strength loss rate by 4.25%. After 30 freeze-thaw cycles, the substitution rate of recycled aggregates has a significant effect on compressive strength. Com-pared with R100%, R0% reduced compressive strength by 15.66%. (Figure 15) (Page 15, Section 4.4, Line 403)

  1. Neglects Environmental Impact Assessment:

The research does not evaluate the environmental impact of using recycled concrete, such as potential benefits in reducing carbon footprint or waste. Including such an assessment would strengthen the argument for its sustainability.

Reference: "Introduction" (p. 1, heading 1)

Thank you for your valuable feedback. We have supplemented this section.

The resource utilization of construction solid waste will reduce the discharge of solid waste, occupy less land area, and also reduce the destructive power on soil. Another benefit of the resource utilization of construction solid waste is the reduction of emissions of gases such as CO2. 25% to 50% of the CO2 emitted during cement manufacturing can be reabsorbed by the resource utilization process of waste concrete. Compared with stacking or landfilling methods, it can reduce N2O emissions by 50%, SO2 emissions by 30%, CO emissions by 28%, and CO2 emissions by 10%. (Page 1, Section 1, Line 42)

5.Inadequate Comparative Analysis:

The article does not provide a robust comparative analysis between recycled concrete and conventional concrete in various environmental conditions. More detailed comparisons would help in better understanding the advantages and disadvantages of recycled concrete.

Reference: "Results and discussion" (p. 10, heading 5)

Thank you for your valuable feedback. We have supplemented this section and compared recycled concrete with ordinary concrete.

For instance, the slump value of C30 concrete without recycled aggregate was 111 mm, while the slump of concrete with a 100% substitution rate was only 90 mm, indicating a notable difference of 21 mm. Compared to regular concrete, recycled concrete experiences a significant reduction in slump. (Page 11, Section 4.1, Line 322)

Compared with ordinary concrete, the rebound value of recycled concrete has decreased, and as the substitution rate of recycled coarse aggregate increases, the re-bound value decreases more. The rebound value of ordinary concrete and concrete with 100% recycled coarse aggregate replacement rate decreased by 5mm compared to each other. (Page 12, Section 4.2, Line 350)

Moreover, after the same freeze-thaw cycle of 30 times, the concrete (R100) completely using recycled aggregates showed more quality loss compared to the concrete (RC) using natural aggregates (Page 14, Section 4.3, Line 384)

After 30 freeze-thaw cycles, the substitution rate of recycled aggregates has a significant effect on compressive strength. Compared with R100%, R0% reduced compressive strength by 15.66%. (Page 15, Section 4.4, Line 412)

Round 2

Reviewer 4 Report

Comments and Suggestions for Authors

I consider your article to be acceptable for publication.

Comments on the Quality of English Language

Moderate